

# Phosphoproteomics analysis of serum from dogs affected with pulmonary hypertension secondary to degenerative mitral valve disease

Siriwan Sakarin[1], Anudep Rungsipipat[2], Sittiruk Roytrakul[3], Janthima Jaresitthikunchai[3], Narumon Phaonakrop[3], Sawanya Charoenlappanit[3], Siriwan Thaisakun[3] and Sirilak Surachetpong[1]

[1] Department of Veterinary Medicine, Faculty of Veterinary Science, Chulalongkorn University, Bangkok, Thailand, Bangkok, Thailand
[2] Center of Excellence for Companion Animal Cancer, Department of Pathology, Faculty of Veterinary Science, Chulalongkorn University, Bangkok, Thailand, Bangkok, Thailand
[3] Functional Proteomics Technology Laboratory, National Center for Genetic Engineering and Biotechnology (BIOTEC), National Science and Technology Development Agency, Pathum Thani, Thailand, Bangkok, Thailand

Corresponding author
Sirilak Surachetpong,
sirilakd27@gmail.com

## ABSTRACT

Pulmonary hypertension (PH), a common complication in dogs affected by degenerative mitral valve disease (DMVD), is a progressive disorder characterized by increased pulmonary arterial pressure (PAP) and pulmonary vascular remodeling. Phosphorylation of proteins, impacting vascular function and cell proliferation, might play a role in the development and progression of PH. Unlike gene or protein studies, phosphoproteomic focuses on active proteins that function as end-target proteins within signaling cascades. Studying phosphorylated proteins can reveal active contributors to PH development. Early diagnosis of PH is crucial for effective management and improved clinical outcomes. This study aimed to identify potential serum biomarkers for diagnosing PH in dogs affected with DMVD using a phosphoproteomic approach. Serum samples were collected from healthy control dogs ($n = 28$), dogs with DMVD ($n = 24$), and dogs with DMVD and PH ($n = 29$). Phosphoproteins were enriched from the serum samples and analyzed using liquid chromatography-tandem mass spectrometry (LC-MS/MS). Data analysis was performed to identify uniquely expressed phosphoproteins in each group and differentially expressed phosphoproteins among groups. Phosphoproteomic analysis revealed nine uniquely expressed phosphoproteins in the serum of dogs in the DMVD+PH group and 15 differentially upregulated phosphoproteins in the DMVD+PH group compared to the DMVD group. The phosphoproteins previously implicated in PH and associated with pulmonary arterial remodeling, including small nuclear ribonucleoprotein G (SNRPG), alpha-2-macroglobulin (A2M), zinc finger and BTB domain containing 42 (ZBTB42), hemopexin (HPX), serotransferrin (TRF) and complement C3 (C3), were focused on. Their unique expression and differential upregulation in the serum of DMVD dogs with PH suggest their potential as biomarkers for PH diagnosis. In conclusion, this phosphoproteomic study identified

![PeerJ]

Subjects  Biochemistry, Veterinary Medicine, Zoology
Keywords  Phosphoproteomics, Pulmonary hypertension, Degenerative mitral valve disease, Dogs, Serum, Proteomics, Biomarkers

uniquely expressed and differentially upregulated phosphoproteins in the serum of DMVD dogs with PH. Further studies are warranted to validate the diagnostic utility of these phosphoproteins.

# INTRODUCTION

Pulmonary hypertension (PH), a common complication in dogs affected by degenerative mitral valve disease (DMVD), has been recognized as an abnormally increased pressure in the pulmonary arteries (*Reinero et al., 2020*). DMVD is characterized by progressive degeneration of the mitral valve, leading to mitral valve regurgitation, enlargement of the left atrium and ventricle, and subsequent congestive heart failure (CHF) (*Borgarelli & Buchanan, 2012*; *Kellihan & Stepien, 2010*). In DMVD, an initial back transmission of increased left atrial pressure to pulmonary capillary causes an early stage of PH, which can be reversible, whereas chronic hypoxia from pulmonary edema can cause pulmonary arterial remodeling, leading to irreversible PH (*Chiavegato et al., 2009*; *Guazzi & Arena, 2010*). Moreover, DMVD dogs with PH have a shorter median survival time compared to those without PH (*Borgarelli et al., 2015*).

According to the American College of Veterinary Internal Medicine (ACVIM) consensus statement guidelines for the diagnosis, classification, treatment, and monitoring of PH in dogs, the diagnosis of PH is based on clinical signs and echocardiographic findings (*Reinero et al., 2020*). The clinical signs suggestive of PH are syncope, respiratory distress, exercise intolerance, and right-sided heart failure (*Reinero et al., 2020*). Echocardiography is primarily used to measure estimated pulmonary arterial pressure (PAP) and identify echocardiographic signs of PH involving the ventricles, pulmonary artery, right atrium, and caudal vena cava (*Reinero et al., 2020*). Although echocardiography is an acceptable method for diagnosing PH in dogs (*Kellihan & Stepien, 2010*) it has some limitations, including the need for expensive ultrasound machines and experienced sonographers. Moreover, pulmonary arterial remodeling, especially vascular medial thickening, may manifest before the detection of elevated PAP (*Delgado et al., 2005*; *Liu et al., 2013*; *Sakarin, Rungsipipat & Surachetpong, 2021*). Several studies indicate that prompt treatment in the early stage of PH can reverse medial thickening of pulmonary arterial walls (*Guazzi & Galiè, 2012*; *Sakao, Tatsumi & Voelkel, 2010*). Due to this concern, diagnosing PH by measuring increased PAP using echocardiography may be delayed, potentially lead to irreversible remodeling of the pulmonary arteries. Therefore, there is still a need for additional diagnostic methods that can detect PH earlier in dogs. The aim is to discover new approaches beyond echocardiography to enhance diagnostic and treatment effectiveness. Circulating biomarkers are currently explored for diagnosing PH in humans (*Banaszkiewicz et al., 2022*). Despite reported associations, none have gained

acceptance for PH diagnosis (*Banaszkiewicz et al., 2022*), and the search for biomarkers continues.

Proteomics, a technique for extensive protein characterization, has been widely used to identify potential biomarkers in diseases, including PH (*Bhosale et al., 2017*; *Zhang et al., 2014*). Despite these efforts, the impact on disease diagnosis, progression assessment, and treatment monitoring remains limited. To address this challenge, a new branch called phosphoproteomics has emerged, concentrating on the phosphoproteome for biomarker discovery (*Mumby & Brekken, 2005*; *Urban, 2022*). Proteins, essential for life, undergo post-translational modifications (PTMs) such as phosphorylation, regulating various cellular processes (*Chen & Kashina, 2021*).

Phosphoproteins, involved in diverse cellular functions and often activated in diseases, have potential as disease biomarkers, potentially offering more specific disease stage information than total proteins (*Ghosh et al., 2022*; *Giorgianni & Beranova-Giorgianni, 2016*). Pulmonary arterial remodeling, particularly medial thickening, is a key characteristic of PH and involves a complex interplay of cellular processes, notably the proliferation of smooth muscle cells (*Guignabert & Dorfmuller, 2013*). Protein phosphorylation plays a crucial role in this process, particularly in regulating signaling pathways that control smooth muscle cell proliferation, such as mitogen-activated protein kinase (MAPK) pathway. Phosphorylation of downstream MAP kinase, followed by the phosphorylation of extracellular signal-regulated kinase (ERK)1/2, facilitates growth signaling and cell proliferation. Dysregulation of phosphorylation events may promote medial thickening (*Crosswhite & Sun, 2014*). Circulating phosphoproteins have been explored as diagnostic biomarkers in human cancers (*Ghosh et al., 2022*). In dogs, a sole study on serum phosphoproteome in Babiosiosis revealed alterations in phosphoproteins (*Galán et al., 2018*). However, there has been no study that has examined circulating phosphoproteins in dogs with PH. This study aimed to assess serum phosphoprotein expression in healthy dogs, dogs with DMVD, and dogs with DMVD and PH.

# MATERIALS AND METHODS

## Animals

The study was designed as a prospective cross-sectional controlled study. Blood samples were collected from client-owned dogs presented as clinical cases at the Small Animal Hospital, Faculty of Veterinary Science, Chulalongkorn University, Thailand. The owners provided informed consent before their dogs were enrolled in the study. The study protocol was approved by the Institutional Animal Care and Use Committee, Faculty of Veterinary Science, Chulalongkorn University (number 1831099). Dogs included in this study were elderly small breed dogs with an age older than 7 years and a weight less than 10 kg. All dogs underwent history taking, physical examination, blood pressure measurement, electrocardiography (ECG), thoracic radiography, echocardiography, and blood collection on the same day.

Dogs were excluded from this study if they had other cardiovascular diseases other than DMVD, including dilated cardiomyopathy, myocarditis, valvular endocarditis, mitral valve dysplasia, mitral valve stenosis or aortic stenosis. Additionally, dogs with pulmonary

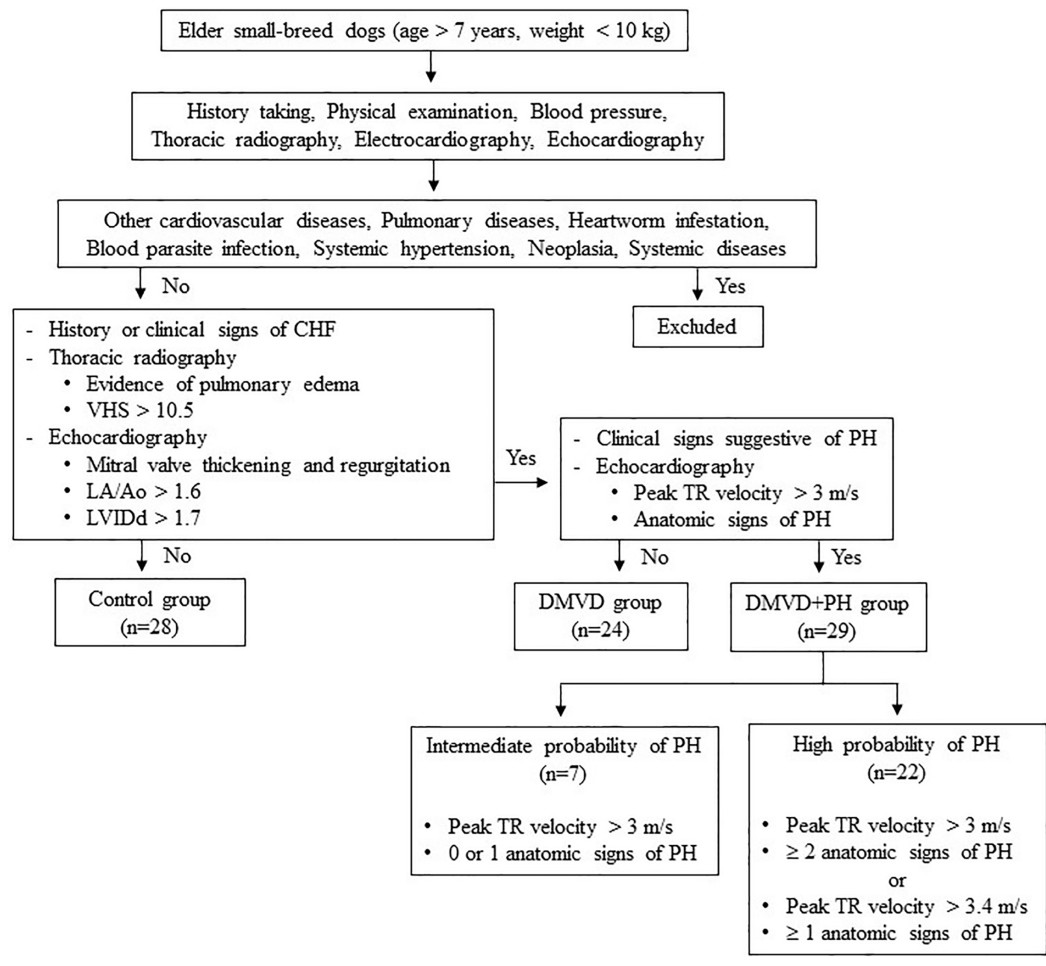

**Figure 1 Inclusion and exclusion criteria for sample selection.**

diseases like chronic obstructive airway disorders or pulmonary parenchymal disease as well as, those with heartworm infestation, blood parasite infection, systemic hypertension, neoplasia, or systemic diseases such as kidney and liver diseases that may cause PH or affect serum protein expression (*Bell et al., 2010*; *Çiftci et al., 2021*; *Escribano et al., 2017*; *Kuleš et al., 2014*; *Munhoz et al., 2012*; *Naseeb et al., 2008*; *Niu et al., 2019*; *Yang et al., 2020*; *Zhou et al., 2017*). Dogs affected with other causes of PH were also excluded. The sample size calculation was derived from a previous study by *Zhou et al. (2012)*, which demonstrated that 12 samples per group were sufficient to detect a two-fold change with a power of 0.8% for 85% proteins. A total of 81 client-owned dogs were enrolled in this study and divided using purposive sampling into three groups: the control ($n = 28$), the DMVD group ($n = 24$), and the DMVD with PH (DMVD+PH) group ($n = 29$), based on the following criteria (Fig. 1).

**The control group** consisted of 28 healthy dogs that had no history or clinical signs of cardiorespiratory disease. Normal heart and lung sounds were detected during the physical examination. Blood pressure was within the normal limit, and no signs of cardiac

arrhythmia were found. To confirm that all dogs enrolled in this group had normal heart and lung conditions, thoracic radiography and echocardiography were performed.

**The DMVD group** included 24 dogs diagnosed with DMVD stage C, exhibiting previous or current signs of CHF as evidenced by clinical signs such as cough, exercise intolerance, or dyspnea, along with radiographic evidence of pulmonary edema. All dogs had radiographic evidence of cardiomegaly, with a vertebral heart score (VHS) greater than 10.5 (*Boswood et al., 2016*), as well as echocardiographic evidence of mitral valve thickening and regurgitation, left atrial enlargement with a left atrial to aorta dimension ratio (LA/Ao) greater than 1.6 and left ventricular enlargement with a normalized left ventricular internal diameter at the end of diastole (LVIDd) greater than 1.7 (*Borgarelli et al., 2015*; *Keene et al., 2019*).

**The DMVD+PH group** comprised 29 dogs with an intermediate to high probability of PH secondary to DMVD stage C. The dogs included to this group were those dogs with PH secondary to DMVD, who had previously undergone echocardiography and were diagnosed with PH and dogs with DMVD undergoing regular monitoring echocardiography every 6 months, who were incidentally discovered to have PH, were included. The probability of PH was determined based on clinical signs suggestive of PH, peak tricuspid regurgitation (TR) velocity, and anatomic signs of PH assessed through history taking, physical examination, and echocardiography. Dogs in this group were classified as having an intermediate probability of PH if their peak TR velocity was greater than 3 m/s with 0 or 1 anatomic sign of PH. Alternatively, they were classified as having a high probability of PH if their peak TR velocity was greater than 3 m/s with ≥2 anatomic signs of PH, or if their peak TR velocity was greater than 3.4 m/s with ≥1 anatomic sign of PH (*Reinero et al., 2020*).

## Sample collection and preparation

The procedures might cause slight pain or distress. To minimize pain or distress in animals, dogs were carefully restrained with experienced veterinary assistants and blood samples were collected using aseptic techniques performed by licensed veterinarians. Dogs were returned to their owners after completion of activity. To minimize pre-analytical variables, a single individual performed all procedures, including sample collection, sample preparation and sample storage according to the following protocol. Three milliliters of blood were collected from the cephalic or saphenous vein of dogs that had been fasted at least 4 h before blood collection to minimize the effect of the meal on serum protein concentration (*Pellis et al., 2012*). The blood samples were then divided, with 1 ml used for complete blood count and blood chemistry analysis and the remaining 2 ml stored in plain Eppendorf tubes. The samples were allowed to clot at room temperature for 2 h. Subsequently, serum samples were separated by centrifugation at 3,000 g for 15 min at 4 °C. Any hemolytic and lipemic serum samples were discarded to minimize their effect on proteomic analysis (*Greco et al., 2017*; *Hsieh et al., 2006*; *Nikolac, 2014*). To prevent protein damage from freeze-thaw cycles, protein degradation, and dephosphorylation, each serum sample was aliquoted and mixed with a protease inhibitor (Halt Protease Inhibitor Cocktail, Thermo Scientific, Waltham, MA, USA) and a phosphatase inhibitor (sodium

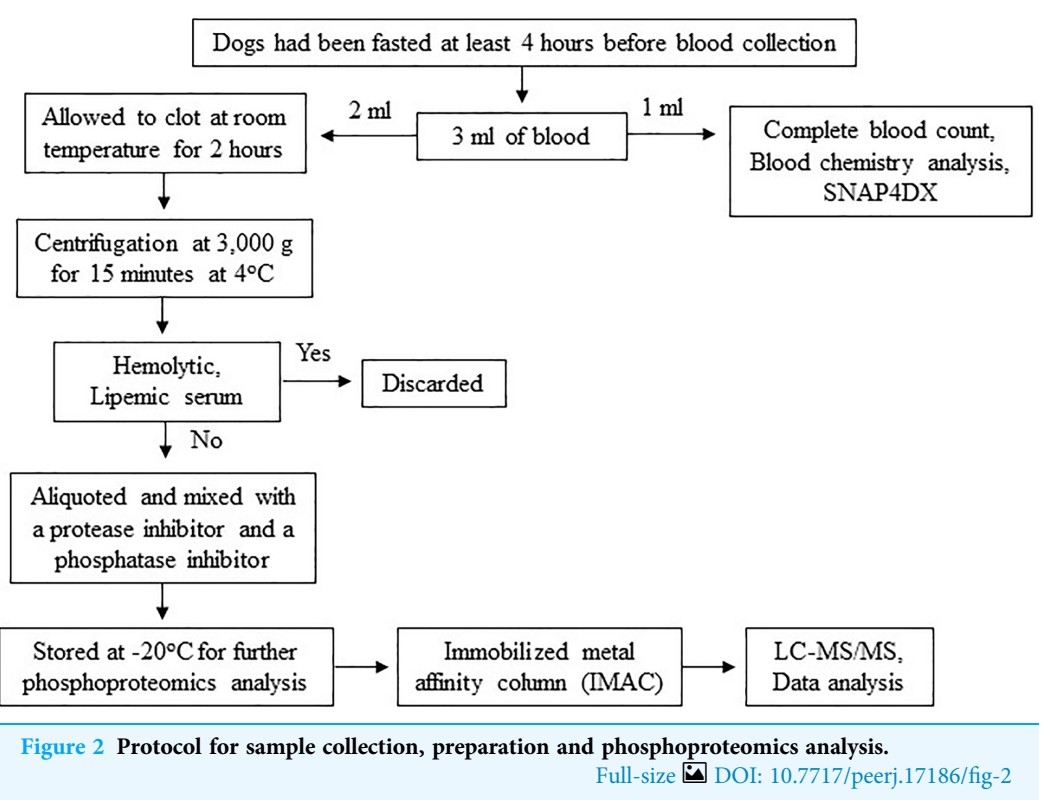

**Figure 2 Protocol for sample collection, preparation and phosphoproteomics analysis.**

orthovanadate: $Na_3OV_4$, Sigma-Aldrich, St. Louis, MO, USA). The samples were then stored at $-20\ °C$ for further proteomic analysis (Fig. 2). All procedures were performed by one person following an identical protocol and under the same circumstances to reduce pre-analytical variables that may affect sensitivity, selectivity, reproducibility, and recovery in obtaining reliable biomarkers (*Greco et al., 2017*; *Luque-Garcia & Neubert, 2007*).

## Analysis of serum phosphoproteins by LC-MS/MS

The aliquoted frozen serum samples were thawed only one time at room temperature before analysis. Lowry's method was used to determine the total protein concentration of each serum sample. Serum phosphoproteins were enriched by an immobilized metal affinity column (IMAC) (Pierce, Thermo Scientific, Waltham, MA, USA) according to the manufacturer's protocol. Based on a previous study indicating that Gallium ion ($Ga^{3+}$) exhibits superior binding specificity for phosphopeptides over nonphosphopeptides (*Machida et al., 2007*), an IMAC resin charged with $Ga^{3+}$ ion was used in this study. Briefly, each serum sample was prepared to a final concentration of 0.5 mg/ml using Lysis/Binding/Wash Buffer. Before use, the column was centrifuged at 1,000 g for 1 min at $4\ °C$ to remove the storage solution. Subsequently, 5 ml of Lysis/Binding/Wash Buffer with CHAPS was added to the column, which was then centrifuged at 1,000 g for 1 min at $4\ °C$ to equilibrate resin. The serum sample was then added to the column, inverted several times to mix, and placed on a platform rocker for 30 min at $4\ °C$. Next, the column was placed in a 50 ml conical tube and centrifuged at 1,000 g for 1 min at $4\ °C$ to collect the flow-through fraction. The resin was washed by adding 5 ml of Lysis/Binding/Wash Buffer

with CHAPS, followed by centrifugation at 1,000 g for 1 min at 4 °C. The washing step was repeated three times. Finally, the phosphoproteins were eluted from the column by adding 1 ml of elution buffer and incubating at room temperature for 3 min. The column was placed into a 50 ml conical tube and centrifuged at 1,000 g for 1 min at 4 °C. This elution step was repeated four times, and the pool fractions were collected for the next step. The phosphoproteins were incubated with 10 mM dithiothreitol (DTT) at 56 °C for 1 h to reduce disulfide bonds. Cysteine residues were alkylated with 100 mM iodoacetamide (IAA) at room temperature in the dark for 1 h, and phosphoproteins were digested with trypsin (Trypsin, Mass Spec Grade, Promega, Madison, WI, USA) for 3 h at room temperature. Finally, 1% formic acid (FA) was added to serum samples to stop enzymatic digestion, and all phosphopeptides were injected into an Ultimate 3000 Nano/Capillary LC System (Thermo Scientific, Waltham, MA, UK) coupled to a ZenoTOF 7600 mass spectrometer (produced by SCIEX, Framingham, MA, USA). Phosphopeptides underwent an enrichment step utilizing a μ-Precolumn (300 μm i.d. × 5mm) packed with C18 PepMap 100 (5 μm, 100 A) (Thermo Scientific, Waltham, MA, USA), and were subsequently separated on a column (75 μm I.D. × 15 cm) filled with Acclaim PepMap RSLC C18 (2 μm, 100Å, nanoViper) (Thermo Scientific, Waltham, MA, USA). The C18 column was maintained at a constant temperature of 60 °C within a column oven. Solvent A and B, containing 0.1% formic acid in water and 0.1% formic acid in 80% acetonitrile respectively, were introduced into the analytical column. A gradient ranging from 5% to 55% solvent B was utilized to elute the phosphopeptides, while maintaining a constant flow rate of 0.30 μl/min over a duration of 30 min. Electrospray ionization was performed at 1.6 kV using the CaptiveSpray system, with nitrogen utilized as the drying gas at a flow rate of approximately 50 l/h. Collision-induced-dissociation (CID) product ion mass spectra were generated using nitrogen as the collision gas. Mass spectra (MS) and MS/MS spectra were acquired in positive-ion mode at a frequency of 2 Hz, covering the range of m/z 150–2,200. The collision energy was tuned to 10 eV in response to the m/z value. For quality control in analytical steps, three replicates of the same sample were analyzed to monitor the reproducibility of the results. Additionally, the digestion of bovine serum albumin served as a quality control sample to assess the performance and reliability of the mass spectrometry instrument and the entire analytical workflow (*Bittremieux et al., 2018*; *Li et al., 2024*; *Vincent et al., 2019*) (Fig. 2).

The individual non-pooled serum samples were analyzed. For protein identification, MaxQuant (version 2.2.0.0) was used to submit the MS/MS spectra to the Andromeda search engine and searched against the *Canis lupus familiaris* UniProt database (*Tyanova, Temu & Cox, 2016*). The significance threshold for protein identification was established with a *p*-value < 0.05 and a false discovery rate (FDR) of 1%. The specific parameters for MaxQuant's standard configuration encompassed allowing a maximum of two missed cleavages, setting the main search mass tolerance at 0.6 daltons, utilizing trypsin as the enzyme for digestion, applying a fixed modification of cysteine through carbamidomethylation, and incorporating variable modifications for methionine oxidation and protein N-terminus acetylation. Peptides were considered for identification and subsequent data analysis if they met the criteria of being at least seven amino acids in

length and containing at least one unique peptide, as outlined in previous studies (*Cottingham, 2009*; *Gupta & Pevzner, 2009*; *Keerapach et al., 2023*). The relationship between differentially expressed phosphoproteins and common cardiovascular drugs was evaluated using the online-based software Stitch (version 5.0) (*Szklarczyk et al., 2016*).

## Statistical analysis

The computer-based software, SPSS (version 22, IBM, Armonk, NY, USA) was used to evaluate the statistically significant difference in the data. The data for all dogs were not subject to blinding. The normality of the data was tested by the Shapiro-Wilk test, and the normally distributed data were expressed as mean and standard deviation (SD). One-way analysis of variance (ANOVA) was used to analyze the differences between the control, DMVD, and DMVD+PH groups, with the Bonferroni test used for *post hoc* analysis. A *p*-value of <0.05 was considered statistically significant. The data of identified serum phosphoproteins were imported to the online-based software Metaboanalyst (version 5.0) for statistical analysis. Partial least squares-discriminate analysis (PLS-DA) was employed to demonstrate the separation between different groups of identified phosphoproteins. An ANOVA test followed by Tukey's *post hoc* test, was performed to identify the differentially expressed phosphoproteins among the groups. A *p*-value and false discovery rate (FDR) of <0.05 was considered statistically significant (*Abooshahab et al., 2020*; *Khan et al., 2019*; *Sajid et al., 2023*). Using MetaboAnalyst, missing values were replaced with half of the minimum value in the dataset to facilitate the continuation of the analysis of the specified phosphoproteins (*Wei et al., 2018*; *Xia et al., 2009*).

## RESULTS

This study included a total of 81 dogs, composed of 28 healthy control dogs, 24 dogs with DMVD stage C, and 29 dogs affected with PH secondary to DMVD stage C. All of the samples were included in the experiment; none were excluded. The normal group consisted of 11 Shih-Tzus, six Chihuahuas, four Poodles, four Pomeranians, and three Yorkshire Terriers. The DMVD group comprised seven Poodles, seven Pomeranians, six Chihuahuas, two Miniature Pinchers, one Shih-Tzu, and one mixed breed. The DMVD +PH group included eight Poodles, eight Chihuahuas, four Shih-Tzus, three Miniature Pinchers, two Pomeranians, two mixed breeds, one Jack Russel Terrier and one Schnauzer. The age of the dogs varied significantly among the groups. The healthy control dogs (9.14 ± 2.16 years) were significantly younger than the dogs in the DMVD (11.33 ± 1.97 years) and DMVD+PH groups (11.97 ± 2.37 years) (*p* < 0.001). However, the body weight of dogs did not differ among the groups (*p* = 0.697). In terms of sex distribution, the control group had an almost equal number of males (12) and females (16), while the DMVD+PH group had 13 males and 16 females. The DMVD group had a higher proportion of male dogs (19) compared to female dogs (five).

According to history taking and physical examination, all dogs in the control group had no history or evidence of cardiovascular and respiratory disease. However, all dogs in the DMVD and DMVD+PH groups exhibited clinical signs of CHF, such as cough, exercise intolerance, and respiratory distress. In the DMVD+PH group, additional clinical signs

suggestive of PH, such as syncope and ascites, were noted. The standard cardiovascular drugs prescribed in this study included angiotensin-converting enzyme inhibitors (ACEIs), furosemide, pimobendan, spironolactone, a combination of amiloride and hydrochlorothiazide (Moduretic®), and sildenafil. The hematologic and blood chemistry profiles of all enrolled dogs were within the normal limit. All dogs were negative for the heartworm antigen test (SNAP4Dx test, IDEXX Laboratories, Inc. Westbrook, Maine, USA). Electrocardiography revealed no evidence of cardiac arrhythmia in any of the dogs. Thoracic radiographic findings of dogs in the control group showed normal results, while dogs in the DMVD and DMVD+PH groups exhibited cardiomegaly with VHS > 10.5, with or without pulmonary edema, at the time of blood collection. In the DMVD+PH group, pulmonary artery enlargement and right-sided heart enlargement were observed. Echocardiography was performed to confirm the cardiac structural changes, and the results showed that all dogs in the control group had unremarkable cardiac abnormalities. Dogs in the DMVD and DMVD+PH groups exhibited left atrial and left ventricular enlargement. In the DMVD+PH group, all dogs had TR velocity greater than 3 m/s and were classified into intermediate ($n = 7$) and high probability of PH ($n = 22$) based on TR velocity and several anatomic signs of PH (Table 1).

## Phosphoprotein identification by LC-MS/MS

After enriching serum phosphoproteins, we identified a total of 1,467 phosphoproteins as illustrated in the heat map (Fig. 3). This study analyzed individual non-pooled serum samples. A total of 1,074 phosphoproteins were identified within distinct groups, with 236 exclusives to two groups, while 157 were present in all three groups. It should be noted that 1,074 phosphoproteins identified in a specific group may not be uniformly found in every dog within that group, resulting in missing values represented as a grey area in the heatmap. Utilizing partial least squares discriminant analysis (PLS-DA), we observed moderate separation between the identified phosphoproteins in the DMVD+PH group and those in the control and DMVD groups, whereas there was an overlapped between the control and DMVD groups (Fig. 4). An ANOVA test with subsequent Tukey's *post hoc* analysis, revealed significant differences ($p < 0.05$) in 42 out of 1,467 identified phosphoproteins among the control, DMVD and DMVD+PH groups (Fig. 5). Further investigation using the online-based software Stitch (version 5.0) examined the interaction between the differentially expressed phosphoproteins and common cardiovascular drugs. This analysis identified four phosphoproteins including albumin, haptoglobin, hemoglobin subunit beta and anionic trypsinogen correlated with commonly used cardiovascular drugs, including enalapril, benazepril, ramipril, furosemide, pimobendan, spironolactone, a combination of amiloride and hydrochlorothiazide (Moduretic®), and sildenafil (Fig. 6). However, the software did not identify the remaining 38 phosphoproteins. Among the 42 differentially expressed phosphoproteins, nine were uniquely expressed in the DMVD+PH group, 31 were upregulated, and two were downregulated compared to the control and DMVD groups. Notably, 15 out of 31 upregulated phosphoproteins in the DMVD+PH group exhibited fold changes greater than two compared to the DMVD group. For further explanation of potential associations

**Table 1 Clinical data of dogs in the control, degenerative mitral valve disease (DMVD) and degenerative mitral valve disease with pulmonary hypertension (DMVD+PH) groups.**

| Parameters | Control ($n$ = 28) | DMVD ($n$ = 24) | DMVD+PH ($n$ =29) | $p$-value |
|---|---|---|---|---|
| **Sex (male/female)** | 12/16 | 19/5 | 13/16 | |
| **Clinical signs** | | | | |
| Cough | – | 24/24 | 29/29 | |
| Exercise intolerance | – | 18/24 | 23/29 | |
| Respiratory distress | – | 6/24 | 8/29 | |
| Syncope | – | – | 5/29 | |
| Ascites | – | – | 7/29 | |
| **Heart sound** | | | | |
| Normal | 28/28 | – | – | |
| Systolic heart murmur | – | 24/24 | 29/29 | |
| **Heart rate (bpm)** | 129.43 ± 16.87 | 133.63 ± 27.04 | 137.10 ± 25.05 | 0.462 |
| **Lung sound** | | | | |
| Normal | 28/28 | 13/24 | 6/29 | |
| Increased | – | 13/24 | 18/29 | |
| Crackle | – | 2/24 | 5/29 | |
| **Blood pressure (mmHg)** | 144.00 ± 18.36 | 138.33 ± 20.25 | 134.24 ± 19.31 | 0.166 |
| Electrocardiography | | | | |
| Respiratory sinus arrythmia | 28/28 | 23/24 | 21/29 | |
| Sinus rhythm | – | 1/24 | 8/29 | |
| **Thoracic radiography** | | | | |
| Vertebral heart score | 9.54 ± 0.77 | 11.67 ± 0.87[a] | 12.21 ± 0.96[a] | <0.001 |
| Pulmonary edema | – | 6/24 | 10/29 | |
| Pulmonary artery enlargement | – | – | 9/29 | |
| Right-sided heart enlargement | – | – | 9/29 | |
| **Echocardiography** | | | | |
| Mitral valve regurgitation | – | 24/24 | 29/29 | |
| LA/Ao | 1.15 ± 0.16 | 2.00 ± 0.41[a] | 2.17 ± 0.60[a] | <0.001 |
| LVIDd | 1.22 ± 0.13 | 1.81 ± 0.26[a] | 1.72 ± 0.43[a] | <0.001 |
| Peak TR velocity (m/s) | – | – | 3.86 ± 0.63 | |
| Right ventricular systolic pressure (mmHg) | – | – | 61.13 ± 21.51 | |
| **Blood profiles** | | | | |
| RBC (x10⁶ cell/μL) | 6.88 ± 0.84 | 6.96 ± 0.78[b] | 6.17 ± 1.14[a,c] | 0.005 |
| WBC (x10³cell/uL) | 8.86 ± 3.26 | 9.75 ± 2.47[b] | 12.60 ± 3.96[a,c] | <0.001 |
| Platelet (x10³cell/μL) | 310.36 ± 90.56 | 332.88 ± 94.21 | 363.90 ± 101.11 | 0.111 |
| ALT (IU/L) | 42.29 ± 16.44 | 67.13 ± 38.15[a] | 63.65 ± 42.78[a] | 0.019 |
| ALP (IU/L) | 57.21 ± 40.46 | 109.29 ± 84.79[a] | 116.62 ± 93.60[a] | 0.009 |
| Creatinine (mg/dL) | 0.89 ± 0.12 | 0.96 ± 0.23 | 0.99 ± 0.20 | 0.112 |
| BUN (mg/dL) | 17.45 ± 6.60 | 33.06 ± 15.62[a] | 38.49 ± 19.59[a] | <0.001 |

**Notes:**
Data are reported as mean ± standard deviation (SD).
The significant difference was assessed by one-way ANOVA at $p < 0.05$.
[a] Significant difference at $p < 0.05$ compared with the control group.
[b,c] Significant difference at $p < 0.05$ comparing between the DMVD and DMVD+PH groups.
LA/Ao, Left atrium to aorta ratio; LVIDd, Left ventricular internal diameter at end diastole; TR, Tricuspid regurgitation; RBC, Red blood cells; WBC, White blood cells; ALT, Alanine aminotransferase; ALP, Alkaline phosphatase; BUN, Blood urea nitrogen.

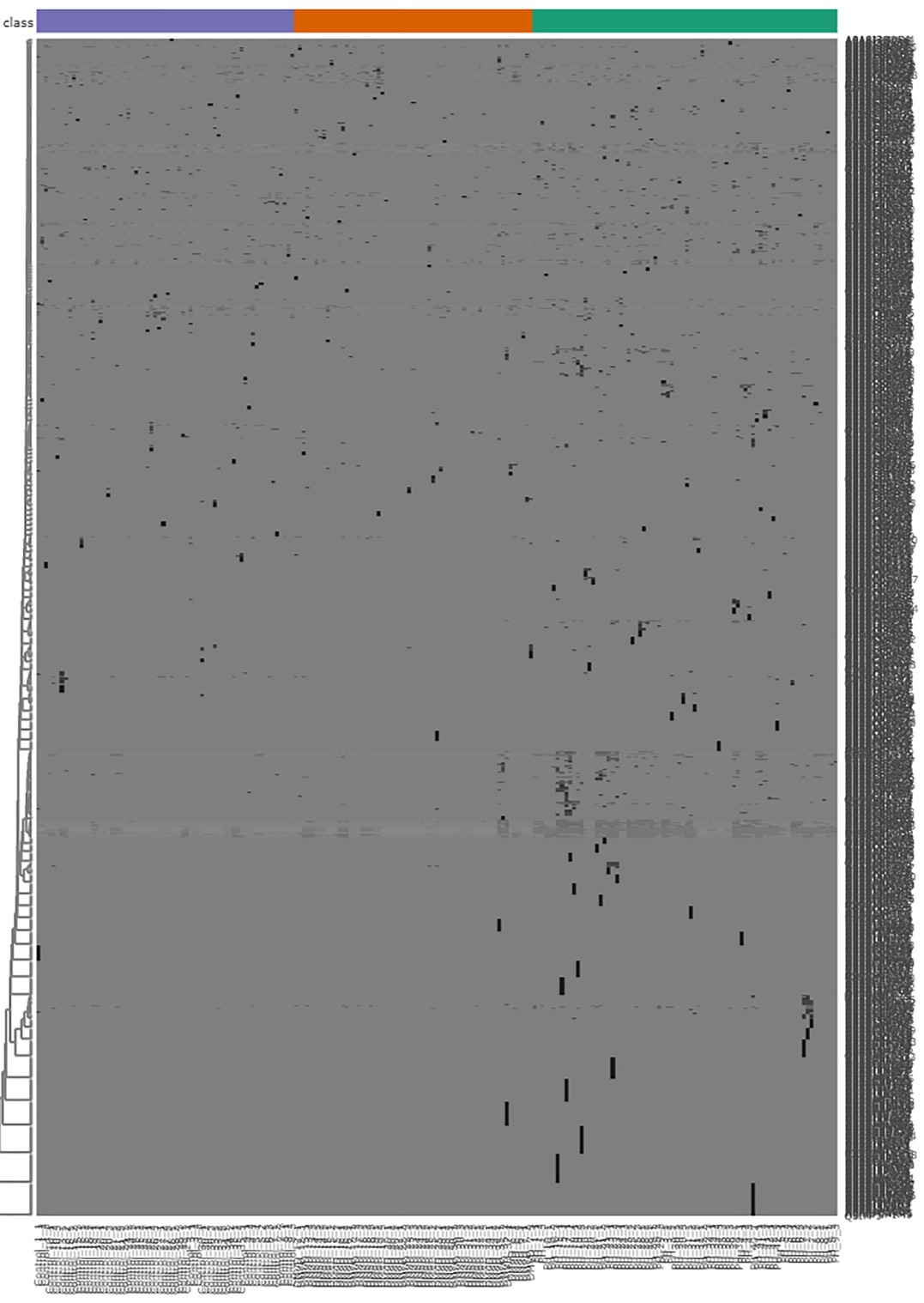

**Figure 3 Heat map of all identified phosphoproteins in the control group, degenerative mitral valve disease (DMVD) group and degenerative mitral valve disease with pulmonary hypertension (DMVD+PH) group.** A total of 1,467 phosphoproteins were identified with different expression levels among groups. Samples are in columns and identified phosphoproteins are in rows. The color indicated phosphoproteins intensity that change from very low (light grey) to extremely high (black). The color scale on the right depicts the range of expression values.

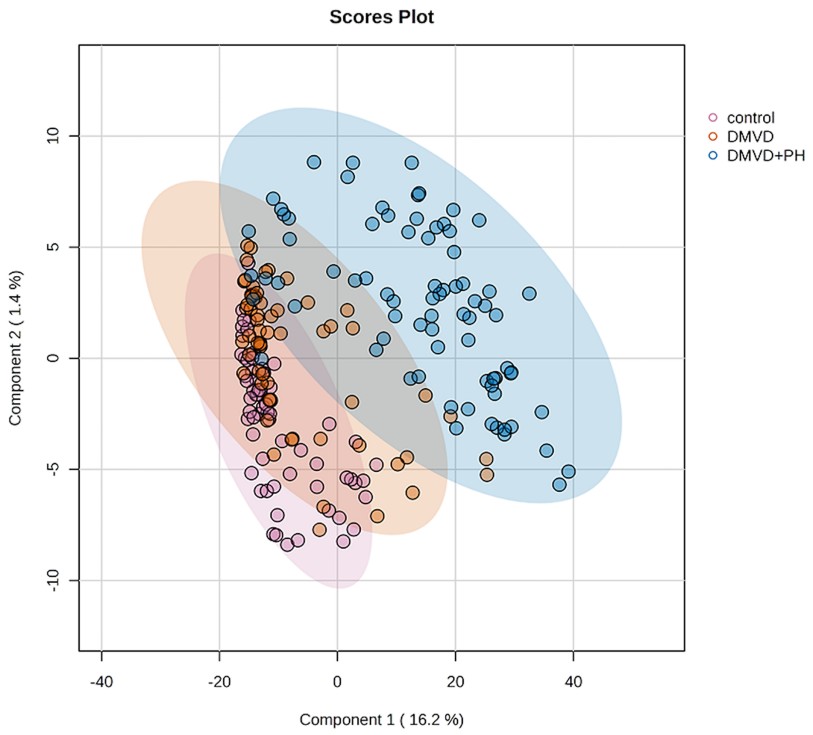

**Figure 4 Partial least squares-discriminant analysis (PLS-DA) of all identified phosphoproteins clustered by groups including the control group, degenerative mitral valve disease (DMVD) group and degenerative mitral valve disease with pulmonary hypertension (DMVD+).** Colored dots represent individual samples and colored areas represent 95% confidence interval.

with PH, we focused on three uniquely expressed phosphoproteins and three upregulated phosphoproteins in the DMVD+PH group: small nuclear ribonucleoprotein G (SNRPG), alpha-2-macroglobulin (A2M), zinc finger and BTB domain containing 42 (ZBTB42), hemopexin (HPX), serotransferrin (TRF) and complement C3 (C3) (Fig. 7). The intensity of these identified phosphoproteins showed no correlation with body weight and PAP ($p >$ 0.05). However, HPX (r = 0.262, $p$ = 0.018) and TRF (r = 0.252, $p$ = 0.023) showed weak correlation with age.

## DISCUSSION

This study aimed to explore the serum phosphoproteome in healthy control dogs, dogs with DMVD, and dogs with DMVD and PH. The key finding of this study was that utilizing phosphoproteins enrichment followed by LC-MS/MS could serve as a method to identify potential phosphoproteins that might act as biomarkers for diagnosing PH in dogs with DMVD. Three uniquely expressed phosphoproteins in the DMVD+PH group: SNRPG, A2M and ZBTB42 and three up-regulated phosphoproteins in the DMVD+PH group: HPX, TRF and C3 were chosen to explain their relationship with PH in dogs. A weak correlation between age and the intensity of HPX and TRF was found, suggesting that these phosphoproteins might not be good candidates for biomarkers of PH in dogs with DMVD. Due to its association with vascular remodeling, as indicated in the literature

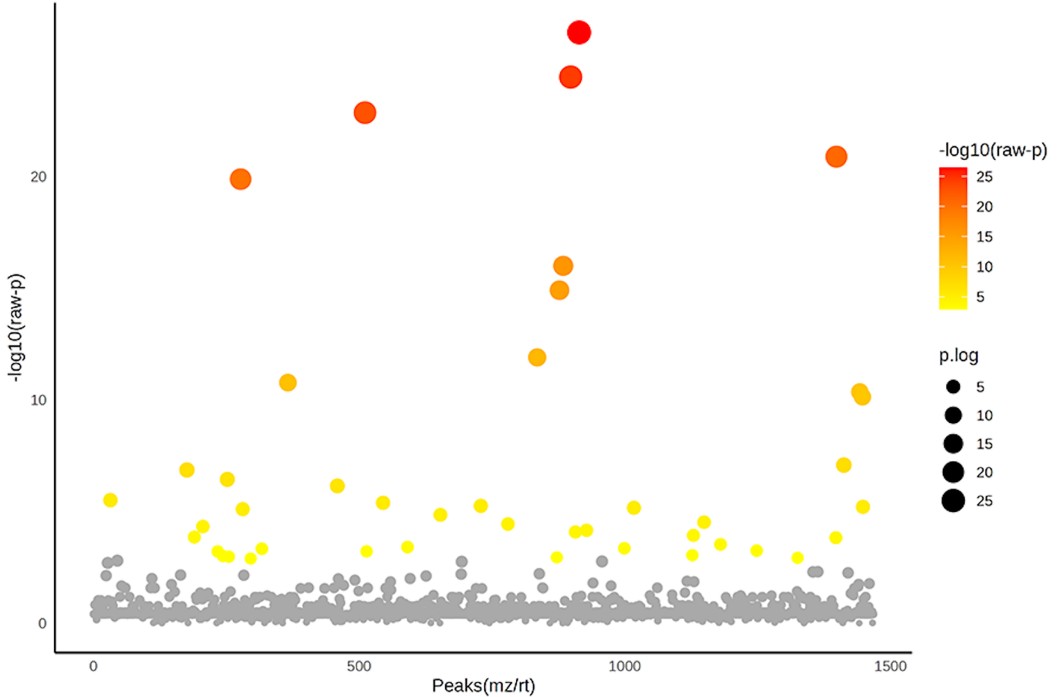

**Figure 5 Analysis of variance (ANOVA) plot of significantly identified phosphoproteins in comparison among the control group, degenerative mitral valve disease (DMVD) group and degenerative mitral valve disease with pulmonary hypertension (DMVD+PH) group.** An ANOVA test revealed 42 out of 1,467 identified phosphoproteins exhibited significant differences among the control, DMVD and DMVD +PH groups. The x-axis represents the phosphoprotein peaks, identified by their mass-to-charge ratio (m/z) and retention time (rt). The y-axis represents the −log10 of the raw $p$-value associated with each peak. Colored dot represents significantly expressed phosphoproteins with $p < 0.05$ while the grey dot represents phosphoproteins without statistical significance.

review, A2M might be a potential biomarker of PH in dogs affected with DMVD. The investigation into phosphorylation changes in serum not only yields potential biomarkers but may also offer insights into specific signaling pathways associated with PH. Therefore, the discovery of these phosphorylation profiles has the potential not only to enhance diagnostic capabilities but also to guide the way for novel therapeutic targets in the future.

Changes in phosphoproteins have been reported in lung tissues and pulmonary arterial smooth muscle cells (PASMCs) of human patients with PH compared to control patients (*Luo et al., 2022*; *Sitapara et al., 2021*). However, no research has been conducted on serum phosphoproteome in PH, either in human patients or dogs. To the best of the author's knowledge, this study represents the first attempt to identify phosphoproteome in the serum of dogs affected by PH secondary to DMVD.

The identification of phosphoproteins as disease biomarkers can be challenging due to their low abundance compared to nonphosphorylated proteins. Therefore, the enrichment of phosphoproteins before analysis is a crucial step (*Zhou et al., 2009*). Various methods can be employed for phosphoproteins enrichment, including immunoprecipitation using phospho-specific antibodies, titanium dioxide ($TiO_2$) chromatography, or immobilized metal ion affinity chromatography (IMAC) (*Delom & Chevet, 2006*; *Fíla & Honys, 2012*; *Zhou et al., 2009*). Immunoprecipitation is typically used to target specific phosphorylated amino acids, making it less suitable for large-scale studies.

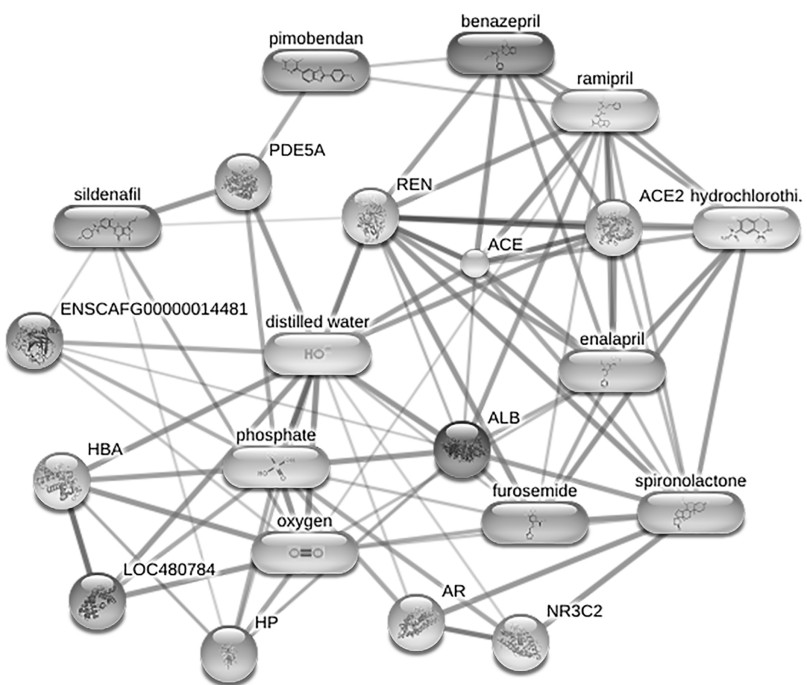

**Figure 6 Network of protein-cardiovascular drugs interaction analyzed by Stitch, version 5.0.** Four phosphoproteins including albumin (ALB), haptoglobin (HP), hemoglobin subunit beta (LOC480784) and anionic trypsinogen (ENSCAFG00000014481) correlated with commonly used cardiovascular drugs, including enalapril, benazepril, ramipril, furosemide, pimobendan, spironolactone, a combination of amiloride and hydrochlorothiazide (Moduretic®), and sildenafil. The differentially expressed phosphoproteins showed a correlation with cardiovascular drugs. The strength of the associations at the functional level was evaluated by edge confidence scores. The strong relationships with high edge confidence scores (>0.700) are presented as thick lines. Abbreviations: ACE, angiotensin-converting enzyme; ALB, albumin; AR, androgen receptor; ENSCAFG00000014481, anionic trypsinogen; HBA, hemoglobin A; HP, haptoglobin; LOC480784, hemoglobin subunit beta; NR3C2, mineralocorticoid receptor; PDE5A, phosphodiesterase 5; ENS REN, renin.

$TiO_2$ chromatography has limitations related to the physical properties of $TiO_2$ and requires the optimization of $TiO_2$-to-peptide ratio. On the other hand, IMAC is widely used for phosphoprotein enrichment as it is commercially available and involves fewer preparation steps (*Delom & Chevet, 2006*; *Fíla & Honys, 2012*; *Jaros et al., 2012*). Several phosphoproteomics studies in serum have successfully employed IMAC as the phosphoprotein enrichment method before proteomic analysis (*Felix et al., 2011*; *Jaros et al., 2012*). In the present study, IMAC was utilized for phosphoprotein enrichment before conducting LC-MS/MS analysis to identify phosphoproteins in serum samples. The results revealed that 1,467 phosphoproteins were detected in the serum samples of dogs in the control, DMVD, and DMVD+PH groups.

Phosphoproteomics is a subset of proteomics that specifically focuses on the identification of phosphorylated proteins, which play a crucial role in various cellular processes occurring during disease stages. Studying phosphoproteins in serum may provide biomarkers that are associated with pathological conditions and more specific than total proteins.

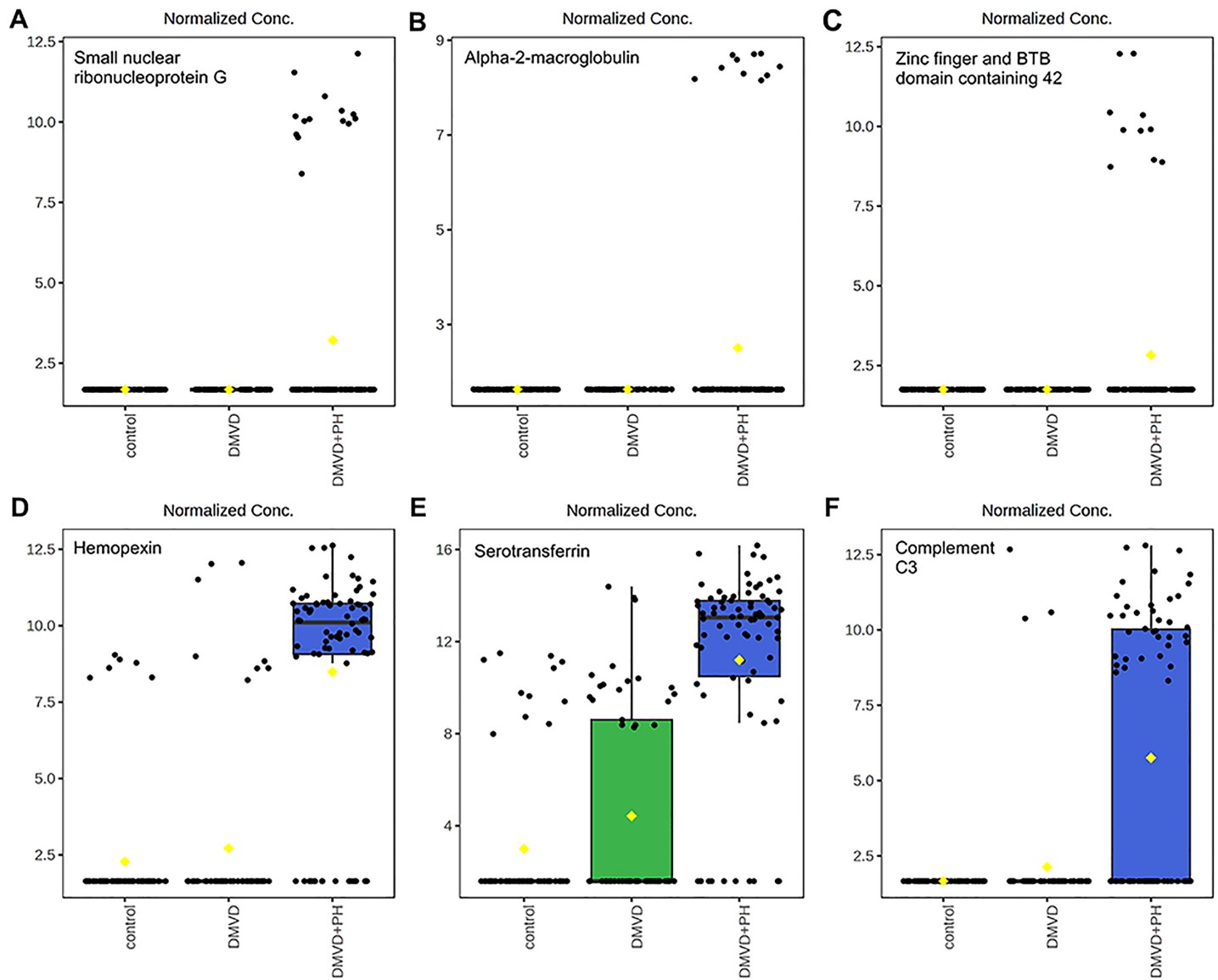

**Figure 7 Boxplot of the focused uniquely expressed phosphoproteins (top row) and upregulated phosphoproteins (bottom row) in the degenerative mitral valve disease with pulmonary hypertension (DMVD+PH) group that have $p$-value < 0.05 and fold changes >2 when comparin.** The focused phosphoproteins such as, small nuclear ribonucleoprotein G (A), Alpha-2-macroglobulin (B), Zinc finger and BTB domain containing 42 (C), were exclusively detected in the serum of dogs affected with PH secondary to DMVD. On the other hand, Hemopexin (D), Serotransferrin (E) and Complement C3 (F) were identified in the serum of DMVD dogs both with and without PH, but their level was significantly upregulated in those with PH. The y-axis represents normalized peak intensity of phosphoproteins obtained in LC-MS/MS analysis. The box ranges from the 25[th] to 75[th] percentiles, with whiskers extending to 1.5 times the interquartile range, and horizontal lines inside each box representing the 50[th] percentile or median. Black dots indicate individual data points, while yellow dot in a boxplot indicates the mean of the data.

This present study focused on identifying uniquely expressed phosphoproteins and upregulated phosphoproteins in the DMVD+PH group. The findings of the study indicated a significant number of phosphoproteins that were exclusively expressed in each group. The selection of potential candidate phosphoproteins took into account their $p$-value and their association with the disease (*Nakayasu et al., 2021*; *Paulovich et al., 2008*;

*Schelli, Zhong & Zhu, 2017*). Based on these criteria, three uniquely expressed phosphoproteins and three upregulated phosphoproteins were chosen for further investigation in the DMVD+PH group: small nuclear ribonucleoprotein G (SNRPG), alpha-2-macroglobulin (A2M), zinc finger and BTB domain containing 42 (ZBTB42), hemopexin (HPX), serotransferrin (TRF) and complement C3 (C3).

According to the proteins functional analysis and pathway search by PANTHER, unfortunately, all focused phosphoproteins were unclassified by PANTHER. Therefore, we reviewed the association of these proteins with PH in previous studies.

Small nuclear ribonucleoprotein G (SNRPG) is a component of the spliceosome, responsible for pre-mRNA splicing to produce mature mRNA, which can be translated into proteins (*Mabonga & Kappo, 2019*). Dysregulation of RNA splicing can lead to abnormal expression and function of proteins involved in various cellular processes, including vascular function and remodeling (*Fei et al., 2016*). Currently, there is no available information about the role of SNRPG in PH. Therefore, further research is necessary to understand the specific role of SNRPG in the pathogenesis of PH.

Alpha-2-macroglobulin (A2M) is a large protein found in biological fluids. It is primarily produced by the liver, but it can also be produced by other cells such as macrophages and fibroblasts (*Lagrange et al., 2022*). A2M plays a significant role in inflammation and infections. It inhibits proteinases released by activated leukocytes or proteinases secreted by invading microorganisms, making it a key player in protecting tissues from damage and preventing excessive inflammation (*Vandooren & Itoh, 2021*). Due to chronic inflammation associated with vascular dysfunction, the interplay of A2M with this process has been studied. It has been reported that A2M plays a role in vascular smooth muscle cell contraction (*Nassar et al., 2002*). Circulating A2M can be used as a biomarker for cerebral small vessel disease in ischemic stroke patients (*Nezu et al., 2013*) and is associated with endothelial dysfunction in patients with chronic stroke or cardiovascular risk factors (*Shimomura et al., 2018*). However, the association of A2M and PH has not been reported. Recently, A2M was found to be present in platelets and platelet activation may increase plasma level of A2M (*Huang et al., 2021*). In both PH human patients and PH-induced animal models, platelet activation has been noted and is known to play a role in pulmonary vascular remodeling (*Hu et al., 2010*; *Varol, Uysal & Ozaydin, 2011*; *Zanjani, 2012*). Very little is known about platelet activation in PH dogs. A study presented at the 2022 ACVIM Forum showed that platelets were hyperactive in dogs with DMVD and PH (*Duler, 2022*). Further investigation is needed to explore the connection between A2M, platelet activation and PH.

Zinc finger and BTB domain containing 42 (ZBTB42) belongs to the zinc finger protein family and functions a transcriptional factor that binds to target DNA sequences and regulate transcription. It is highly expressed in human and mouse skeletal muscle, localizing in the nucleus of skeletal muscle fibers, and plays a role in skeletal muscle development (*Devaney et al., 2011*). Modifications in a variety of transcriptional factors have been identified as important regulators in PH and associated with pulmonary vascular remodeling (*Yang et al., 2023*). However, the role of ZBTB42 in pathogenesis of PH has not been explored.
Hemopexin (HPX) is glycoprotein found in the blood and helps maintain the balance of heme levels in the bloodstream and acts as an antioxidant to protect injured tissues from oxidative damage (*Tolosano & Altruda, 2002*). Its synthesis is stimulated following inflammation (*Liang et al., 2009*) or when heme concentration is high (*Garland et al., 2016*). Hemolysis and cell-free hemoglobin have reportedly been linked to the pathology of PH (*Meegan et al., 2023*; *Rafikova et al., 2018*). Red blood cell lysis in PH might be caused by the remodeling of the pulmonary microvascular system. In PH human patients, circulating cell-free hemoglobin was increased compared to the healthy controls and was associated with pulmonary vascular resistance and PAP. Additionally, HPX levels were also elevated in PH patients, suggesting mechanisms that may provide protection from the harmful effect of cell-free hemoglobin (*Meegan et al., 2023*).

Serotransferrin (TRF), also known as serum transferrin, is a glycoprotein found in the blood that plays a crucial role in iron metabolism. It binds to iron in circulation and facilitates its transport into cells. Additionally, TRF regulates hepcidin, a peptide hormone produced in the liver responsible for controlling the distribution of iron throughout the body (*Gkouvatsos, Papanikolaou & Pantopoulos, 2012*). While no direct association of TRF with PH has been reported, iron deficiency and hepcidin, regulated by TRF, are directly implicated in the development of PH (*Quatredeniers et al., 2021*). Intracellular iron deficiency in pulmonary arterial smooth muscle cells (PASMCs) can induce pulmonary vasoconstriction, ultimately leading to PH and right heart failure in mice (*Lakhal-Littleton et al., 2019*). Hepcidin contributed to pulmonary vascular remodeling by enhancing human PASMCs proliferation (*Ramakrishnan et al., 2018*). In PH human patients, reduced serum iron, elevated TRF and hepcidin levels were observed (*Rhodes et al., 2011*; *Robinson et al., 2014*).

Complement component 3 (C3) is a protein in the innate immune system that plays a role in the complement cascade, a series of reactions that act as the first line of defense to eliminate pathogens and injurious stimuli (*Markiewski & Lambris, 2007*). C3 has been suggested to be involved in the pathogenesis of PH, as its expression was found to be increased in the lungs of patients with idiopathic pulmonary arterial hypertension (IPAH) and hypoxia-induced PH mice. Additionally, the loss of C3 attenuated pulmonary arterial remodeling in hypoxia-induced PH mice (*Bauer et al., 2011*). Moreover, circulating C3 has been implicated as a diagnostic biomarker in IPAH (*Abdul-Salam et al., 2006*; *Zhang et al., 2009*).

Among the selected phosphoproteins including SNRPG, A2M, ZBTB42, HPX, TRF and C3, only A2M and TRF have been studied in relation to vascular remodeling. The intensity of none of these phosphoproteins correlated with estimated PAP accessed by echocardiography. The intensity of TRF weakly correlated with age, whereas A2M did not, suggesting that A2M might be a possible candidate for diagnosing PH in dogs affected with DMVD. Nevertheless, the specific role of this phosphoprotein in PH has not been reported. Further research is necessary to understand the specific role of A2M in the pathogenesis of PH.

## Limitations

Several limitations in this study should be acknowledged. Firstly, it was challenging to completely control for various factors such as age, sex, breed, and previous treatment with cardiovascular drugs, which could potentially influence the expression of serum phosphoproteins. Dogs with varying ages, sexes, and breeds exhibit physiological differences that may influence serum phosphoprotein expression. While age has been reported to impact the total protein concentrations, sex and breed generally do not, but specific protein levels may be influenced. In humans, A2M and TRF levels vary with age and sex (*Han et al., 2014*; *Higgins, Chan & Adeli, 2017*; *Tungtrongchitr et al., 2003*); however, our study demonstrated that only the intensity of phosphoTRF correlated with age. According to the effect of cardiovascular drugs, the online-based software Stitch (version 5.0) revealed no association between the drugs and the focused phosphoproteins. Secondly, the unequal distribution of each group results from a limited time to collect samples. Proteomic analysis needs all samples processed simultaneously, requiring collection from all groups before the experiment. Prolonged storage risks sample deterioration. Therefore, we need to collect samples as equitably as possible among all groups. Thirdly, the phosphoproteomic techniques utilized in this study require expensive equipment and skilled scientists, making their routine use in veterinary clinical practice currently unfeasible. Despite these practical limitations, this study offers valuable insights into phosphoproteomic research in veterinary medicine. The findings of this study can serve as a foundation for future research endeavors aimed at developing a novel diagnostic method for PH in dogs with DMVD.

## CONCLUSIONS

In conclusion, this study aimed to identify unique and upregulated serum phosphoproteins expressed in the DMVD+PH group using phosphoproteins enrichment followed by LC-MS/MS. The results demonstrate the potential of this technique for identifying phosphoproteins biomarkers for diagnosing of PH secondary to DMVD in dogs. Among the uniquely expressed and differentially upregulated phosphoproteins, alpha-2-macroglobulin (A2M) might be a potential candidate for diagnosing PH in dogs affected with DMVD based on their *p*-value and the evidence suggested to be associated with the pathogenesis of pulmonary arterial remodeling. The findings of this study can serve as a foundation for future research endeavors aimed at developing a novel diagnostic method for PH in dogs with DMVD.

## ACKNOWLEDGEMENTS

The authors would like to thank all staff at the Small Animal Hospital, Faculty of Veterinary Science, Chulalongkorn University, for their help in sample collection and all staff at Functional Proteomics Technology Laboratory, National Center for Genetic Engineering and Biotechnology (BIOTEC), National Science and Technology Development Agency for laboratory facility support.

### Funding

The study was funded by the 90th Anniversary of Chulalongkorn University Fund (Ratchadaphiseksomphot Endowment Fund) and the 100th Anniversary Chulalongkorn University Fund for Doctoral Scholarship. The funders had no role in study design, data collection and analysis, decision to publish, or preparation of the manuscript.

### Grant Disclosures

The following grant information was disclosed by the authors:
90th Anniversary of Chulalongkorn University Fund (Ratchadaphiseksomphot Endowment Fund).
100th Anniversary Chulalongkorn University Fund for Doctoral Scholarship.

### Competing Interests

The authors declare that they have no competing interests.

### Author Contributions

- Siriwan Sakarin performed the experiments, analyzed the data, prepared figures and/or tables, authored or reviewed drafts of the article, and approved the final draft.
- Anudep Rungsipipat analyzed the data, authored or reviewed drafts of the article, contributed reagents and materials, and approved the final draft.
- Sittiruk Roytrakul analyzed the data, prepared figures and/or tables, authored or reviewed drafts of the article, and approved the final draft.
- Janthima Jaresitthikunchai performed the experiments, prepared figures and/or tables, and approved the final draft.
- Narumon Phaonakrop performed the experiments, prepared figures and/or tables, and approved the final draft.
- Sawanya Charoenlappanit performed the experiments, prepared figures and/or tables, and approved the final draft.
- Siriwan Thaisakun performed the experiments, prepared figures and/or tables, and approved the final draft.
- Sirilak Surachetpong conceived and designed the experiments, prepared figures and/or tables, authored or reviewed drafts of the article, and approved the final draft.

### Animal Ethics

The following information was supplied relating to ethical approvals (*i.e.*, approving body and any reference numbers):

The Institutional Animal Care and Use Committee, Faculty of Veterinary Science, Chulalongkorn University.

### DNA Deposition

The following information was supplied regarding the deposition of DNA sequences:

The raw data from MaxQuant is available at the ProteomeXchange Consortium *via* the jPOST partner repository: JPST002998; and at ProteomeXchange DOI 10.6019/PXD050923.

## Data Availability

The raw data from MaxQuant is available at the ProteomeXchange Consortium (DOI 10.6019/PXD050923) *via* the jPOST partner repository: JPST002998.

https://repository.jpostdb.org/entry/JPST002998

## Supplemental Information

Supplemental information for this article can be found online at http://dx.doi.org/10.7717/peerj.17186#supplemental-information.

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
