# Peer review of "Phosphoproteomics analysis of serum from dogs affected with pulmonary hypertension secondary to degenerative mitral valve disease"

_PeerJ, doi:10.7717/peerj.17186_

## Round 0.1 · original submission · Major Revisions

This study is very interesting and useful for canine cardiology. However, a weak point of the study is that this manuscript was part of a previous study that has not been published. Most of the important data was not included in this manuscript. Some sentences are unacceptable for publication, and the major finding was not well synthesized. I strongly recommend that the author needs to explain more clearly and should not refer to unpublished data. If you would like to wait for your unpublished manuscript, I recommend withdrawing this manuscript until your referred manuscript is published. We do not assume that you are engaging in 'salami publication,' so please avoid it.

This information is unaccented; you need to revise and show all related data.

Line 234 “This study used serum samples from a previous study, and all information about the dogs has been reported elsewhere (data under review for publication).”

Line 323; This study showed that the identified phosphoproteins in serum samples were more strongly associated with diseases or pathological conditions than identified proteins from proteomics study, which were associated with several systems and less specificity (data under review for publication).

Reviewer 1 ·

Basic reporting

This topic is very interesting in veterinary medicine. This article investigates the serum Phosphoproteomic in pulmonary hypertension dogs secondary to MMVD compared to healthy dogs. However, the study's weak point is most of the important data was not included in this manuscript. Some sentences are unacceptable for publication, and the major finding was not well synthesized.

Abstract
1. Please add a brief sentence: How is phosphoproteomic related to PH or DMVD?
2. Please add a brief sentence about the phosphoproteomic study. Why is this study interested in phosphoproteins instead of mRNA expression or total protein expression?

Introduction
3. In line 73. Please add a citation for this sentence: “According to the American College of Veterinary Internal Medicine (ACVIM) consensus statement guidelines for the diagnosis, classification, treatment, and monitoring of PH in dogs, the diagnosis of PH is based on clinical signs and echocardiographic findings.”
4. In line 75. Please add a citation for this sentence: “The clinical signs suggestive of PH are syncope, respiratory distress, exercise intolerance, and right-sided heart failure.”
5. In line 80. Please change “Medial Thickening” to “Vascular Medial Thickening”
6. In line 82, please add the paragraph mentioning PH and the biomarker for PH as well as its limitation for detecting PH.
7. In lines 83-115, the sentence was too redundant. The author should start with Proteomics, the relationship of proteomic and PH, the research question, and then the aim of the study and hypothesis. Please revise this section.

Experimental design

Material and Methods
8. In lines 124-127, the author mentions, "All dogs underwent history taking, physical examination, blood pressure measurement, electrocardiography (ECG), thoracic radiography, echocardiography, and blood collection on the same day.” Please provide these data in the manuscript.
9. In line 128, please add more detail about the term “other cardiovascular diseases.” Please clarify and add more detail to the material and method.
10. Please clarify respiratory diseases that may cause PH. How about “chronic obstructive airway disorders” and “lung parenchymal disease”? Does the author exclude these diseases? Please add more detail to the material and method.
11. In line 130, what are systemic diseases that may cause PH or affect serum protein expression? Please clarify and add more detail to the material and method.
12. In lines 1361-137, why is each group's number of dogs unequal?
13. Does the author rule out pulmonary hypertension
14. In lines 154-159, the author mentioned that “Dogs in this group were classified as having an intermediate probability of PH if their peak TR velocity was greater than 3 m/s with 0 or 1 anatomic sign of PH. Alternatively, they were classified as having a high probability of PH if their peak TR velocity was greater than 3 m/s with > 2 anatomic signs of PH or if their peak TR velocity was greater than 3.4 m/s with > 1 anatomic sign of PH.” Regarding this paragraph, why the severity of PH in “The DMVD+PH group” is not equal? The probability of PH should be consistent and put in Figure 1. Please clarify and add more detail in the material and method.

Validity of the findings

Result
13. In lines 234-235, Please clarify this unprofessional sentence “This study used serum samples from a previous study, and all information about the dogs has been reported elsewhere (data under review for publication).”
14. In lines 243-268, Please provide the table data of physical examination, blood pressure measurement, electrocardiography (ECG), thoracic radiography, echocardiography, and blood profiles in the manuscript. The author cannot describe without the raw data.

Discussion
15. What is the major finding of this study?
16. In line 326; this sentence is unacceptable “data under review for publication”. Please delete it.

the major finding was not well synthesized.

Additional comments

Figure and figure legend
17. Figure 1: the figure is low quality. Please revise it.
18. Figure 1: Please add more inclusion criteria for PH in the figure.
19. Figure 2: the figure is low quality. Please revise it.
20. Figures 3-7 are very low quality. The figure legend lacked the details. The abbreviation was not clarified. This figure cannot stand alone by itself and is hard to understand.

Reviewer 2 ·

Basic reporting

This is an in vivo study that sought to identify potential serum biomarkers for diagnosing pulmonary hypertension (PH) in dogs with degenerative mitral valve disease (DMVD) using a phosphoproteomic approach. The study included a total of 81 dogs, composed of 28 healthy control dogs, 24 dogs with DMVD stage C, and 29 dogs affected with PH secondary to DMVD stage C. The authors reported that 9 uniquely expressed phosphoproteins were increased in the serum of dogs in the DMVD+PH group and 15 differentially upregulated phosphoproteins were detected in DMVD+PH group compared to the DMVD group. Additionally, the authors suggested that Alpha-2-macroglobulin (A2M) and Serotransferrin (TRF) may be the possible candidate for diagnosing PH in dogs affected with DMVD.

Major concerns:

As the authors mentioned that age, sex, breed, and previous treatment with cardiovascular drugs, which could potentially influence the expression of serum phosphoproteins. Thus, the authors should discuss the potential effects of these factors on 3 uniquely expressed and 3 upregulated phosphoproteins reported in this study, especially A2M and TRF.

Experimental design

The information about the dogs should be provided.

Validity of the findings

Although this is an interesting study, the authors did not provide the clear mechanism underlying the association between these upregulated phosphoproteins and the pathogenesis of PH in this study.

Additional comments

Minor concerns:

1. The words, “(data under review for publication)”, should be replace with the words “(unpublished data)”

2. Figure 3 (inset in right top corner), “DMVD+” should be “DMVD+PH”

3. Figure 4 (legend), “(DMVD+” should be “(DMVD+PH)

Reviewer 3 ·

Basic reporting

The labels of Figure 5 and Figure 7 are unclear, rendering these figures unreadable. For example, what does the x-axis of Figure 5 represent? Additionally, for Figure 7, please clarify the meaning of the yellow dot and black dots. Clear labeling will improve the interpretability of these figures.

Experimental design

no comment

Validity of the findings

1. In line 243, the authors state that ages are significantly different between the healthy control group and the groups with DMVD and DMVD with PH. When conducting statistical analysis, such covariates may need consideration. Could the authors elaborate on how these covariates were addressed?

2. The reproducibility of the proteomics workflow is crucial for biomarker discovery. How is the overall technical performance of the workflow? Were quality control samples included when running the experimental samples? Including data from quality control samples would significantly enhance the overall quality of the manuscript.

3. Regarding Figure 3, does the grey area indicate missing values in the sample cohort? If so, a high level of missing values could compromise the power of statistical analysis. Could the authors explain how missing values were addressed during data processing?

4. To validate potential biomarkers, it would be valuable to assess the correlation between identified changed proteins and traditional diagnostic methods, such as PTH-related measurements using echocardiography.

5. According to Figure 7, the intra-group variation appears to be substantial. Could the selected cutoff for determining significantly changed proteins result in a high rate of false discovery?

---

## Round 0.2 · Major Revisions

Thank you very much for your revised manuscript following the comments mentioned by our potential reviewers in the first round. However, after reviewing your revised manuscript, there are some important points that you need to address to enhance the quality of your article. Many points raised by the reviewers are very important, and I agree with their feedback that you need to add the requested information and correct the data. Please carefully read and respond to all comments point by point.

Reviewer 1 ·

Basic reporting

No comment

Experimental design

No comment

Validity of the findings

Please revise Table 1
- the number of dogs in each topic was very confusing. Please put "n" after the number.
- In the Echocardiographic parameter, please add the data of right ventricular systolic pressure (RVSP) in this topic.

Additional comments

Thank you for addressing previously made comments. Although the main objections made by the authors were mostly well addressed, there are still some important remaining issues that – from my point of view – should first be addressed before a publication of this manuscript can be recommended.

Again, Thank you for addressing all comments. I feel this almost mitigates my initial objections to publication. Good luck with your further research.

Reviewer 2 ·

Basic reporting

The authors have satisfactorily addressed most of my concerns. The quality of the manuscript improved a lot.

Experimental design

I recommended acceptance after minor revision.

- Table 1, number and sex of animals (male/female) in each group should be provided.

Validity of the findings

- Table 1, number and sex of animals (male/female) in each group should be provided.

Reviewer 3 ·

Basic reporting

1. In line 84, the author mentions that vascular medial thickening may manifest before the detection of elevated PAP. Based on the information provided in the introduction, it is unclear how vascular medial thickening is related to the challenges in the diagnosis of PH in dogs.
2. Line 86, please add the reference for the sentence ‘Circulating biomarkers are currently explored for diagnosing PH in humans.’
3. In paragraph 2, the author intended to justify why phosphoproteomics was chosen in study dog PH, but provided insufficient evidence. For example, 'Circulating phosphoproteins have been explored as diagnostic biomarkers in human cancers'. However, protein biomarkers have also been approved by FDA for cancer diagnosis in human (e.g. OVA1 test). 'In dogs, a sole study on serum phosphoproteome in Babiosiosis revealed alterations in phosphoproteins', which is irrelevant with PH. Instead, the justification on why using phosphoproteomics would benefit more from briefly discussing the potential relationship between protein phosphorylation and the pathology of PH.
4. In line 103, the authors inferred the potential significance of the study for human PH. However, for scientific rigor, since the study presented in the manuscript is limited to dogs and does not include human subjects, it is inappropriate to suggest any relationships with humans in the study questions."
5. Missing raw data. Please upload the raw files of LC-MS analysis to public repository, and add a Data Availability section.
6. Missing raw data. Please include output file of MaxQuant as supplementary files.

Experimental design

1. Missing experimental details. Please provide the detailed LC conditions and MS settings.
2. Missing experimental details. Line 214, the description of database searching is vague. Please provide detailed search parameters.

Validity of the findings

1. Line 238-244. What is the purpose of describing dog breeds in each group? Please provide a conclusion related with this information.
2. Line 275-279, the use of "certain", "some" and "others" are not scientific expression, please use actual percentage and numbers when describing the results.
3. The data interpretation in the manuscript is insufficient and weak. While the common practices in analyzing phosphoproteomics data, such as pathway analysis, gene ontology analysis, protein-protein interactions analysis, are not present in the manuscript, the authors only examined the interaction between the differentially expressed phosphoproteins and common cardiovascular drugs, which seems unusual. The authors need to justify why only such analysis was made.
4. The major findings in the manuscript were not highlighted.

---

## Round 0.3 · accepted · Accept

We are thrilled to inform you that your article has been accepted to publish in our journal soon! We cannot express enough how grateful we are for the tremendous effort you put into revising the manuscript based on the comments provided by our editor and reviewer teams. We truly believe that your contribution will make a significant impact on the advancement of knowledge in the field.

As you continue your journey of academic excellence, we hope that you will consider our journal as your first choice for any future submissions.

Warmest regards,

Reviewer 1 ·

Basic reporting

no comment

Experimental design

no comment

Validity of the findings

no comment

Reviewer 2 ·

Basic reporting

The authors have satisfactorily addressed my comments.

Experimental design

No comment

Validity of the findings

No comment